# Scarring Alopecias: Pathology and an Update on Digital Developments

**DOI:** 10.3390/biomedicines9121755

**Published:** 2021-11-24

**Authors:** Donna M. Cummins, Iskander H. Chaudhry, Matthew Harries

**Affiliations:** 1The Dermatology Centre, Salford Royal Hospital, Northern Care Alliance NHS Foundation Trust, Manchester M6 8HD, UK; donna.cummins@nca.nhs.uk; 2Department of Pathology, Royal Liverpool University Hospital, Liverpool L7 8XP, UK; iskander.chaudhry@nhs.net; 3Centre for Dermatology Research, University of Manchester, MAHSC and NIHR Manchester, Biomedical Research Centre, Manchester M13 9WU, UK

**Keywords:** scarring alopecia, diagnosis, hair follicle, digital pathology, artificial intelligence, review, lichen planopilaris, frontal fibrosing alopecia, discoid lupus erythematosus, pseudopelade of brocq, central centrifugal cicatricial alopecia, folliculitis decalvans

## Abstract

Primary cicatricial alopecias (PCA) represent a challenging group of disorders that result in irreversible hair loss from the destruction and fibrosis of hair follicles. Scalp skin biopsies are considered essential in investigating these conditions. Unfortunately, the recognised complexity of histopathologic interpretation is compounded by inadequate sampling and inappropriate laboratory processing. By sharing our successes in developing the communication pathway between the clinician, laboratory and histopathologist, we hope to mitigate some of the difficulties that can arise in managing these conditions. We provide insight from clinical and pathology practice into how diagnoses are derived and the key histological features observed across the most common PCAs seen in practice. Additionally, we highlight the opportunities that have emerged with advances in digital pathology and how these technologies may be used to develop clinicopathological relationships, improve working practices, enhance remote learning, reduce inefficiencies, optimise diagnostic yield, and harness the potential of artificial intelligence (AI).

## 1. Introduction

The management of hair disorders can be challenging. The key to successful treatment is securing a robust diagnosis. Unfortunately, these conditions frequently pose diagnostic challenges to both dermatologists and pathologists alike, with misdiagnosis risking exposing patients to inappropriate or futile treatment strategies. Scalp skin biopsies can be a useful tool in identifying the underlying pathological process, help guide management, and ultimately establishes realistic treatment goals. Appropriate biopsy site selection along with close communication between the clinician and histopathologist will improve the chance of an accurate diagnosis and is considered advantageous in determining the procedural approach and management plan for these complex patients [1,2].

In this review we present the key histopathological features of the most common scarring alopecias, with an emphasis on securing an accurate diagnosis. We provide guidance on biopsy site selection, and how tissue sampling and laboratory processing can be optimised to help differentiate between different scarring hair loss types. We advocate the use of digital pathology to improve the communication between the clinician and pathologist, including the use of virtual clinicopathological (CPC) meetings. Together, we hope this review will provide a pathway to the successful management of these challenging cases and explore how advances in such technology could transform diagnostic histopathology.

## 2. Cicatricial Alopecias

Cicatricial (scarring) alopecias are an uncommon group of permanent patchy hair loss conditions clinically characterised by the loss of visible follicular ostia, associated with variable degrees of visible inflammation. Histologically, the main feature is inflammatory destruction of the hair follicle epithelium with replacement by scar-like fibrous tissue. They are broadly divided into primary or secondary cicatricial alopecias (CA) depending on the target of the pathological process. In Primary CA (PCA), inflammation specifically targets the hair follicle epithelium, whereas in secondary CA the hair follicle is damaged as part of a wider destructive process within the skin (e.g., radiation, thermal burns, etc.)—see Nanda et al. [3]. Permanent alopecia may also occur in traditionally non-scarring alopecia disorders, where follicular drop out occurs in long-standing cases. Here the fibrous tract seen may be subtle and easily overlooked but histologically is reminiscent of follicular scarring seen in PCA. This “biphasic” presentation can be seen in longstanding cases of androgenetic alopecia, alopecia areata and traction alopecia [4].

## 3. Scalp Biopsy, Site Selection and Optimizing Sample Quality

Due to overlapping and changing clinical signs, securing a diagnosis on clinical features alone can often be challenging in PCA. Thus, scalp biopsy is regarded as the key investigation of any suspected cases of scarring alopecia [5].

To optimise the diagnostic yield, scalp biopsies should be taken from an affected area where the hair follicles are clinically inflamed and reduced in number, but still present (NB. usually toward the edge of the patch). Trichoscopy can help to identify subtle signs of inflammation and guide biopsy site selection [6]. Samples should be taken using a 4 mm punch biopsy orientated parallel to the angle of the hair shaft growth (to prevent cross-cutting the HFs) and deep enough to include the entire HF within the sample (i.e., into subcutis) [7]. It is important to recognise that the punch biopsy device used can influence adequacy of specimen size as the internal diameter of some punch instruments can render a smaller specimen than anticipated [8]. Ideally two biopsies should be obtained for horizontal and vertical sections and sent in separate cassettes [9]. Although some advocate a triple biopsy technique to reduce the risk of inconclusive findings and a further procedure, this will result in greater scarring and (in the authors opinion) is usually not required with careful biopsy site selection [10].

If only a single biopsy is obtained, either vertical or horizontal sectioning can be adopted. The advantage of horizontal sectioning is that this technique results in visualization of all the follicles sampled, allowing for an appreciation of the diagnostic features even if only focally present within the specimen. In contrast, vertical sections offer a more complete morphological overview of the follicle. Specifically, inflammatory changes at the dermato-epidermal junction (DEJ) can be identified precisely; an attribute not found inhorizontal sections. However, in vertical sectioning, sampling errors occur more frequently as only a limited number of hair follicles within each sample are usually visualised; this limitation can be at least partially mitigated against by taking further levels if necessary. Interestingly, a study of 76 cases comparing both approaches [11] favoured vertical sectioning for diagnosing scarring alopecias when only one method was available.

One option when tissue is limited is to take three vertical sections from a single punch and then re-embed for horizontal sectioning, on the understanding this results in a smaller sample. As the experience of assessing horizontal sections varies, discussion with the local pathologist about the preferred option may be helpful.

## 4. Sample Processing and Pathology Reporting

Many techniques exist in the laboratory for processing alopecia specimens. In our experience, it is preferable that bisecting of horizontal biopsies is not performed at the time of cut up, but rather carried out during embedding by experienced skilled staff. Horizontal samples are bisected and both halves are embedded, cut surface down, with red ink used to mark the cut surfaces (Figure 1).

It is important that the epidermis and the subcutaneous fat is included in both the horizontal and vertical sections and this requires education and training of laboratory staff.

Vertical sections may show between 2–3 follicular units in a 4mm punch biopsy and, as hair grows at an angle, tangential follicular cuts are frequently produced. However, a good biopsy technique (see above), along with appropriate sample orientation at the embedding stage, can reduce this risk. Although vertical sections do not allow for the interpretation of follicular density or hair counts (to assess hair cycle phases and miniaturisation), key attributes essential in making the diagnosis of a non-scarring pathology [1], these hair cycle parameters are less important in PCA assessment as the hair density seen is entirely dependent on the biopsy site selected. Six levels are usually adequate for vertical sections and twelve levels are typically required for horizontal sections to study all areas of the skin from the epidermis through to subcutis.

Special stains are available as a panel and can be used in line with experience, with H&E, EVG and PAS in standard use. Elastin staining (e.g., Verhoeff-Van Gieson (VVG)) on vertical specimens allows for a distinction between normal dermis and scarred areas in later stage scarring, with scarred areas lacking elastic fibres [12]. Further, distinct patterns of elastin loss are reported in different PCA entities and may help differentiate the underlying cause in later stage disease [13]. Thickened basement membrane zone, apoptotic cells and fungal elements are highlighted by Periodic Acid Schiff (PAS) staining [14,15]. Toluidine blue can help to visualise premature desquamation of the inner root sheath (IRS), helpful for the diagnosis of central centrifugal cicatricial alopecia (CCCA)—see below [16]. Moreover, toludine blue and colloidal iron are used to highlight dermal mucin, a useful marker for distinguishing cutaneous lupus erythematosus from lichen planopilaris. Direct immunofluorescence studies (DIF) is of particular value where cutaneous lupus erythematosus and autoimmune blistering conditions are considered in the differential diagnosis. It is important that careful attention is paid to site selection with DIF sampling taking place at the periphery of an active site of inflammation [2].

## 5. PCA Classification (and Its Limitations)

In 2001, The North American Hair Research Society (NAHRS) proposed a classification of PCA based on the predominant inflammatory cell infiltrate seen on histology (i.e., lymphocytic, neutrophilic, mixed) [17]. This publication was an important step in PCA management, as accurate classification is vital for appropriate disease description, epidemiological data collection and research participation. Whilst the NAHRS groupings help catalogue the different PCA conditions when coupled with clinical findings, they do have limitations. In particular, clinical and histological features of PCAs frequently overlap between distinct entities and may also change over time within the same person [18]. Unfortunately, the absence of specific molecular markers currently makes it difficult to form distinctions between PCA entities beyond clinical and histological description [7,19,20,21].

Furthermore, attempts to attribute unique characteristics to each PCA to improve histological categorisation has broadly been unsuccessful [22]. A blinded histopathological study found that whilst lymphocytic and neutrophilic groups can readily be distinguished histologically, clinically distinct PCA within these groups could not. The authors suggest that the striking similarity of the histologic changes observed between the entities indicates a common pathway phenomenon converging on end-stage scarring changes characterized by complete follicular destruction, replacement with fibrous tissue and absence of an inflammatory infiltrate [23].

### 5.1. PCA Pathobiology

The pathogenesis of cicatricial alopecia remains speculative but the collapse of hair follicle (HF) bulge immune privilege, damage to epithelial hair follicle stem cells (eHFSCs), and epithelial-mesenchymal transition (EMT) of these cells, are likely to be central processes in PCA development of any type [24,25]. Differentiating different PCA subtypes is histologically challenging, particularly in late-stage disease, and may suggest a “final common pathway” in scarring development, akin to the end-stage “cirrhosis” in liver diseases [23]. Additional factors, such as the cytotoxic T cell response, HF microbiome and genetic predisposition likely determine the diverging phenotypes between the different PCA entities [26,27,28]. This has been illustrated by comparing LPP and frontal fibrosing alopecia (FFA); two clinically distinctive (yet histologically identical) entities which appear to display both shared and divergent pathogenesis, likened to the trunk and branches, respectively, of the same “pathogenesis tree” [24].

### 5.2. Epithelial HF Stem Cell Damage

The site of peri-follicular inflammation is instrumental in understanding the pathogenesis of PCAs. In scarring alopecias, the inflammatory infiltrate is located around the bulge region and distal follicle with relative sparing of the proximal bulb. Epithelial HFSCs located in the bulge region (at the insertion point of the arrector pili muscle in the outermost layer of the outer root sheath) are responsible for follicular cycling and ongoing hair viability. Thus, eHFSC damage results in irreversible HF loss and abrogation of the hair cycle. Expression of the eHFSC markers keratin 15 and CD200 appears greatly diminished in some PCAs [27]. Further, permanent HF loss seen in mice, resulting from targeted deletion of keratin 15+ bulge cells, confirm their importance in maintaining HF viability [29]. Whilst the driving force of the peri-follicular inflammation in PCA is unclear, it appears sufficient in destroying eHFSCs and in doing so, the regenerative capacity of the HF [24].

### 5.3. Bulge Immune Privilege Collapse

Shielding healthy HFs from inflammatory cell infiltrate and subsequent damage is provided by a series of mechanisms that restrict antigen presentation from the HF epithelium and suppress immune responses within the tissue. These mechanisms are termed immune privilege (IP) and are present at the bulge throughout the hair cycle and in the bulb during the anagen phase [30,31]. Pro-inflammatory cytokines, particularly interferon (INF)-γ, drive increased expression of IP collapse indicators (MHC class I and class II; β2-microglobulin) and down-regulation of locally generated key immunosuppressants (CD200 and TGFβ2) indicating collapse of the physiological HF IP; thereby exposing (usually hidden) HF antigens to immune surveillance [24]. It is unclear whether IP collapse is a primary event in disease development or just a secondary consequence of the (already established) inflammatory process [24,32].

### 5.4. Epithelial-Mesenchymal Transition

Epithelial-mesenchymal transition (EMT) is the process by which epithelial cells lose polarity and cell-to-cell contact and acquire a mesenchymal phenotype [33]. Expression of an EMT marker SNAI1 has been seen in the fibrotic dermis of FFA and suggests a role of EMT in its pathogenesis [31]. Increased gene and protein expression of mesenchymal markers (e.g., vimentin, fibronectin, CDH2) and reduction in the epithelial marker E-cadherin has been shown in lesional LPP HFs, suggesting that the EMT of bulge HF stem cells plays a pivotal role in the fibrotic response seen in this condition [34], likely driven by TGFβ signalling [35,36].

## 6. Histopathology of PCAs

Histologic features common to all scarring alopecia include loss of sebaceous glands, thinning & asymmetry of the follicular epithelium, perifollicular inflammation, naked hair shafts, premature desquamation of the inner root sheath and, as the condition progresses, follicular or diffuse dermal scarring. Photo-mnemonics have been described as a helpful educational adjunct to illustrate the findings of both scarring and non-scarring alopecias. Relevant to lymphocytic PCA identification, the grouping of two follicles have been described as “eye-like” or “goggle-like” structures on a background of decreased follicular density with follicular scars. These “eyes” represent the fusion of the connective tissue sheaths of adjacent follicles, and the “goggles” represent the outer root sheaths of two follicles surrounded by concentric fibrosis [37] (Figure 2).

Histological features of all PCA are presented in Table 1. Below we focus on the commonest PCAs seen in practice and provide the greatest challenges to diagnosis.

### 6.1. Lichen Planopilaris

Lichen planopilaris is the commonest form of scarring alopecia and includes the variant FFA [7,38]. In classic LPP, erythematous and hyperkeratotic follicles are often found in a zone of inflammation at the hair-bearing periphery of a patch of scarring alopecia. In contrast, FFA is characterised by frontotemporal hairline recession of both terminal and vellus hairs. When LPP is suspected, the biopsy site should be performed where there are retained but reduced hair follicles, identifying the presence of follicular inflammation and particularly perifollicular scale.

Histologically, the interfollicular epidermis is commonly spared in LPP/FFA, but when it is involved, it appears attenuated with flattening or saw toothing of the rete ridges and evidence of pigment incontinence. In early stages, the perifollicular lichenoid infiltrate is mainly localised around the infundibulum and isthmus with evidence of damage to the basal follicular epithelium (“squamatisation” of the basal cell layer); although the classic upper dermal band-like infiltrate seen in cutaneous lichen planus is not a feature of LPP. Lichenoid interface dermatitis is a key distinguishing feature of this condition and can be appreciated between the follicular epithelium and adjacent dermis (Figure 3).

Blurring (but not thickening) of the basement membrane occurs and sebaceous glands are lost early in the disease process [39] and apoptotic keratinocytes (Civatte bodies) may be seen in the distal HF epithelium. Advanced cases show diffuse scarring with a few residual follicular units (Figure 4).

Eventually, hair follicles are completely replaced by fibrous tissue. Elastin staining typically shows a wedge-shaped zone of loss of the elastic fibre network in the location of the destroyed hair follicle, best examined on vertical sections [13] (Figure 5).

Currently, histological findings cannot distinguish between LPP and FFA [23]; however, certain characteristics reportedly differ between the two entities: FFA shows more prominent apoptosis, less inflammation, deeper inflammation and sparing of the interfollicular epidermis, compared with LPP [40,41]. Further, the “follicular triad”, describing the simultaneous involvement of terminal, intermediate and vellus follicles in FFA may suggest an increased susceptibility of miniaturized hairs to the disease process. Thus, early loss of vellus hairs in the frontal hairline in FFA is a useful clinical clue to diagnosis, helping to differentiate FFA from androgenetic alopecia, whilst providing insight into the pathological process that result in this presentation [42,43]. To date, macrophage numbers and polarisation have been the only reported factor found to be significantly different between LPP and FFA, although further work is required to confirm these preliminary results [44].

The major histological differential diagnoses for LPP/FFA are CCCA, chronic cutaneous lupus erythematosus (CCLE) and occasionally folliculitis decalvans (FD). The distinguishing features between these conditions are presented in Table 2 [45].

### 6.2. Chronic Cutaneous Lupus Erythematous

The most common type of CCLE is discoid lupus erythematosus (DLE) with scalp involvement occurring in 60% of patients [46,47]. Clinically, discoid lesions are associated with localised hair loss and inflammation in the centre of the patch. The key clinical features of DLE can be remembered with the acronym “PASTE” (follicular Plugging, skin Atrophy, Scale, Telangectasia and Erythema).

Biopsies should be taken from an inflamed site with residual follicles. When DLE is suspected a DIF specimen should also be obtained from lesional skin [48]. Dermal oedema, pigment incontinence and loss of sebaceous lobules are associated with early disease. Typical histological features include vacuolar interface change with apoptotic keratinocytes along the HF basal layer and at the dermal-epidermal junction between follicles. In contrast with LPP, the lymphocytic infiltrate in DLE involves both the superficial and deep dermis, extending along blood vessels and adnexal structure. The basement membrane is usually thickened, which can be highlighted with a PAS staining. The presence of mucin and an infiltrate of plasma cells in a perivascular or peri-adnexal location strongly supports the diagnosis of DLE.

Two histologic follicular patterns of chronic cutaneous lupus erythematosus in transverse sections have been described: an alopecia areata (AA) like pattern and an LPP-like pattern [49]. Investigators found decreased follicle size, increased number of catagen/telogen hair follicles, pigment casts, and deep perifollicular inflammation, which resemble AA in some samples, whereas perifollicular lamellar fibrosis and lymphocytic inflammation at the level of isthmus and infundibulum reminiscent of LPP were also observed, potentially resulting in a misdiagnosis of LPP. This pitfall can be circumvented by exercising close clinicopathological correlation and identifying the differentiating features of DLE over other PCAs (i.e., thickening of the basement membrane zone, the presence of deep perivascular inflammation, and interstitial mucin).

### 6.3. Central Centrifugal Cicatricial Alopecia

This slowly progressive symmetrical cicatricial alopecia develops over the vertex scalp. It is mostly found in women of African descent and signs of clinical inflammation are varied but uncommon. Disease activity expands in a centrifugal fashion, with diffuse scarring, indistinct margins and retained (and often minaturised) hairs within the patch. Dermoscopy can guide biopsy site selection, and provides the best diagnostic yield when broken hairs, or hair surrounded by a peri-pilar white-grey halo, are sampled [6,16].

On horizontal sectioning eccentric epithelial atrophy is seen. In early disease, lymphocytic infiltration is focused on the distal follicle with premature desquamation of the internal root sheath (PDIRS) frequently observed. Normally, the inner root sheath desquamates within the mid to upper isthmus, whereas in PDIRS, desquamation occurs deep to the isthmus at the dermal/subcutaneous junction. However, PDIRS is not distinctive for CCCA and may be found in any condition where follicles are significantly inflamed. In late-stage disease, destruction of the follicular epithelium occurs with eventual replacement of HFs by follicular scars. Polytrichia and foreign body multinucleate giant cells surrounding free hair shaft fragments are also features of late-stage disease.

A helpful pearl in diagnosing CCCA is the presence of PDIRS in non-inflamed follicles. In one study, PDIRS was identified in 100% of cases relating to active LPP, DLE, acne keloidalis nuchae, and alopecia areata. Whereas, PDIRS in non-inflamed HFs occurred in 73% of CCCA, but only 33% of psoriatic alopecia, 11% of folliculitis decalvans, and 1% of androgenetic alopecia cases, with none of the non-inflamed follicles in LPP or DLE demonstrating this feature [50].

### 6.4. Classic Pseudopelade (Brocq)

Pseudopelade of Brocq (PsB) is an idiopathic cicatricial alopecia that presents with small, circular, non-inflamed and slightly atrophic patches of alopecia on the scalp, with their appearance classically likened to “footprints in the snow”. Follicular scale and erythema are minimal or absent. In 1986, Braun-Falco et al. proposed diagnostic criteria basis of clinical, histopathological, and immunofluorescence features [51]. Unfortunately, the diagnosis of PsB as a distinct entity remains controversial; Amato et al. found that 66% of the cases initially classified as PsB using Braun-Falco’s clinical criteria had diagnostic features of LPP or DLE on scalp histology, suggesting that the majority of cases actually represents a non-inflammatory version of these conditions [52]. Confusingly, the term “pseudopelade” is also frequently used in scarring alopecia to describe any non-inflammatory cicatricial alopecia [52].

As mentioned, the histology in PsB is non-specific, with minimal inflammation seen around the distal follicle and bulge. Atrophy of the follicular epithelium is seen, but no interface changes are associated with the presence of concentric perifollicular and lamellar fibrosis. Cylindrical columns of connective tissue form at the site of former follicles with the development of follicular stelae.

### 6.5. Folliculitis Decalvans

Folliculitis decalvans is a neutrophilic PCA that typically presents as a highly inflammatory cicatricial alopecia. Follicular papules, hair tufting, marked crust and prominent follicular pustules at the advancing hair loss margin are characteristic [7]. Involvement of the vertex and occipital scalp is usual, although the beard area and nape may also be affected. Scalp biopsy should be taken from the active margin of the alopecia patch.

In early disease histology may demonstrate HF epithelial squamatisation with a dense neutrophilic infiltrate most marked around the distal HF, particularly at the infundibulum.

Subsequently, sebaceous glands are destroyed, follicular damage ensues and HF rupture attracts macrophages and foreign body giant cells around destroyed HF in the dermis [7]. Over time, neutrophil numbers reduce, with increased prominence of lymphocytes and particularly plasma cells in later disease (Figure 6).

The loss of elastic tissue throughout the dermis can be extensive [13,53]. Fungal causes should be considered in all cases of neutrophilic alopecias with special stains (Grocott methenamine silver or PAS) performed combined with bedside cultures. Staphylococcus aureus is frequently isolated from lesional scalp tissue in FD, and probably contributes to the disease phenotype [54,55]. Swabs should be taken to identify resistant bacteria and direct therapy [56].

Interestingly, a biphasic presentation of FD and LPP has recently been described where a phenotypic switch between the two disorders appears to occur and clinicians should be aware of this [18]. Further, occasionally referred patients with a histological diagnosis of FD are seen who have no evidence of scarring but features compatible with bacterial folliculitis only. It appears that biopsies of a scalp pustule from these patients show many features histologically for FD. Therefore, close clinicopathological correlation is required when clinical and histological features do not align.

## 7. Role of CPC

The alopecia CPC meeting is a vital component of any hair loss diagnostic service. The process comprises presentation of key clinical feature, review of clinical (+/− trichoscopic) images and discussion of the histology features using a digital slide viewer for each case. Both the clinician and the pathologist can guide the discussion, highlight important clinical and histological features, and allow exploration of the main differential diagnoses within a multi-disciplinary team setting. Advantages of a digital CPC include ability to review clinical images and histology remotely, allowing wider participation despite time or geographical restrictions. Further, feedback from each case will allow for refinement in management practices and can be used as an educational resource to improve standards.

## 8. Digital Pathology—A Promising Frontier

Although pathology technologies have been slow in coming to market, there is a wide range of innovative technologies that are both in the pipeline or sitting on a shelf ready to be utilised. From scanning hardware to interpreting and reporting software, the capabilities of these technologies are growing.

Traditionally, diagnostic histopathology has depended on reviewing glass slides at a microscope. The physical constraint of glass slides has many drawbacks, including risk of loss or damage and delays in obtaining specialist review due to transport issues. In contrast, once glass slides have been scanned and digitized, these images can be rapidly shared with other specialist pathologists, trainees, clinicians, and students anywhere in the world. Thus, computer-based review of high-resolution image files (or ‘virtual slides’) is therefore fast becoming the standard means of primary diagnostics, and now plays a major role in pathology teaching and tissue-based research. Despite these advances, DP has lagged behind its sister speciality Radiology in terms of implementing digital support into practice.

Central to DP is Whole Slide Imaging (WSI), Slides are scanned and converted to digital file that can be viewed on specialised software with quick and easy retrieval at future timepoints at different locations, as required (Figure 7).

This technology allows for high-speed and high-resolution digital analysis of microscopic slides on a computer monitor. Benefits of WSI include a side-by-side comparison of slides, which is not possible with light microscopy. Further, pathologists are no longer tied to a microscope or require physical slides to report cases. Images can be easily shared allowing discussion of complex cases with national and international experts, reducing time delays. However, DP also has its challenges with WSI having longer average review times when compared to light microscopy [57,58], and optimal management of these digital files is yet to be determined, as files produced are typically large (1–4 GB) and there is no common operating format (in contrast to the DICOM format used in Radiology) [59,60]).

## 9. Digital Pathology and Hair Loss

The potential to develop digital pathology to improve specialist hair loss services is significant. Patients are usually referred to a tertiary referral alopecia service for diagnostic advice, guidance on treatment options or to access specialist therapeutics. It is not uncommon for proffered diagnoses to be revised or additional investigations requested to secure a diagnosis. At the Salford Royal NHS Foundation Trust Hair Clinic, we found the referred (original) diagnosis was revised in 24% cases*, and 25% attendees ultimately submitting to scalp biopsy to secure their diagnosis [* n = 251 new patients; January–November 2018; MH unpublished data].

Integration of the digital era heralds exciting opportunities in the study and management of hair disorders. Off-site pathology with a centralised laboratory securely scanning slides reduces transport and storage costs, whilst expediting the results pathway for both clinicians and patients. Digital infrastructure bundles the slides, clinical information, and photographic images to be accessed in a compact online format for case assessment. For the histopathologist, remote ergonomic working becomes a reality, which forges links with pathology and clinical colleagues locally, regionally, and worldwide. Our proposed pathway for digital pathology end-to-end reporting at Salford Royal is demonstrated in Figure 8. Central to any quality improvement initiative is assessing the effectiveness of an intervention. We recently presented our experience of introducing a digital model of histological assessment into routine specialist hair clinical practice [61].

As hair histopathology is a highly specialist field, digital advances create an educational network of expertise to share and discuss challenging and interesting cases. As discussed, CPC is essential for accurate diagnosis; with digital pathology geographical and time barriers are removed, enabling flexible virtual clinicopathological meetings to suit participants. The benefits of digital pathology, and current infrastructure limitations, has been starkly brought into focus with the rapid and dramatic forced adjustment to working practices triggered by the global pandemic of COVID-19.

## 10. Artificial Intelligence Interfaces

A subset of AI which has been of particular interest to pathology is deep learning (DL). Central to this is building a knowledge base for machines to learn. Pattern recognition and diagnostic algorithms could be created to support the pathologist and reduce reporting time; an approach already showing promise in breast cancer quantification of hormone receptor status to support histological assessment in this disease [62]. Thus, these technologies represent further opportunities for education, treatment, and service development.

### Artificial Intelligence and Hair Diseases

The emerging digital technologies highlighted above may also advance AI by establishing infrastructure and data collections that can support computer learning. WSI allows pathologists to annotate the images on the software, highlighting the key diagnostic features on the slide, such as measuring sample size and hair follicle diameter. The ability to identify hair follicles and measure the hair shaft diameter will be advantageous in determining the vellus: terminal ratio. This feature could be extended in the future using deep learning algorithms to generate the terminal to vellus ratio and total hair count per mm2 which would be far more accurate than the current assessment.

## 11. Conclusions

The histological findings of many forms of hair loss are subtle, with accurate diagnosis dependent on distinguishing abnormal from normal follicular architecture. Early diagnosis of scarring alopecia is aided by correct sampling and biopsy technique, but with clinicopathological correlation critical in ultimately determining the underlying condition. Here we show how digital pathology can aid this process and represents a powerful tool for clinical and research data generation going forward.

## Figures and Tables

**Figure 1 biomedicines-09-01755-f001:**
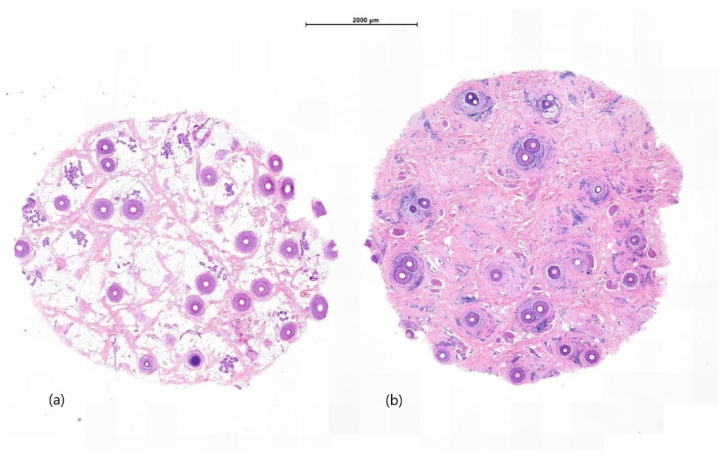
Bisecting of horizontal biopsies. Horizontal sections (**a**) represents a section through subcutaneous fat with embedded hair follicles (**b**) evidence of perifollicular lichenoid inflammation with accompanying scarring (H&E ×5).

**Figure 2 biomedicines-09-01755-f002:**
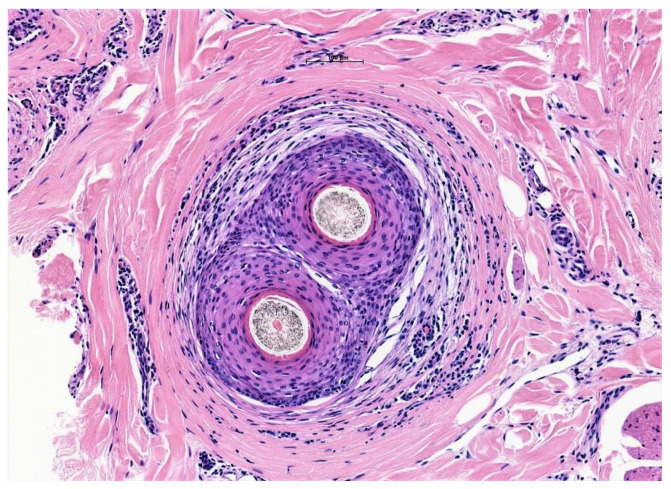
“Eyes” or “Goggles” of LPP. A pair of compound follicular structures with perifollicular inflammation and concentric lamellar fibrosis resembling goggles in a case of LPP (H&E ×200).

**Figure 3 biomedicines-09-01755-f003:**
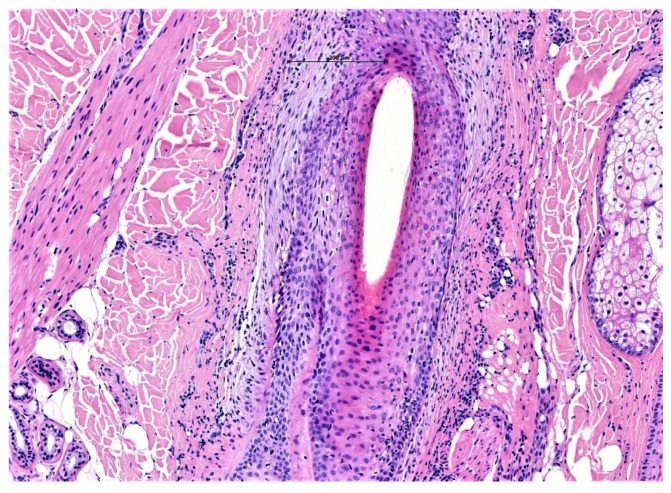
Lichen planopilaris. Medium power magnification showing a vertical section of LPP with lichenoid interface dermatitis involving the infundibulum with perifollicular fibrosis (H&E ×40).

**Figure 4 biomedicines-09-01755-f004:**
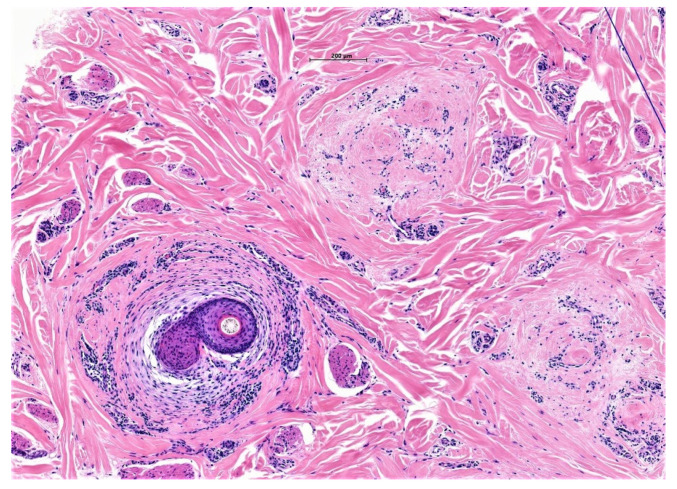
Advanced scarring alopecia. Medium power of a horizontal section showing scarring alopecia with complete loss of follicular units and a residual inflammed follicular structure (H&E ×30).

**Figure 5 biomedicines-09-01755-f005:**
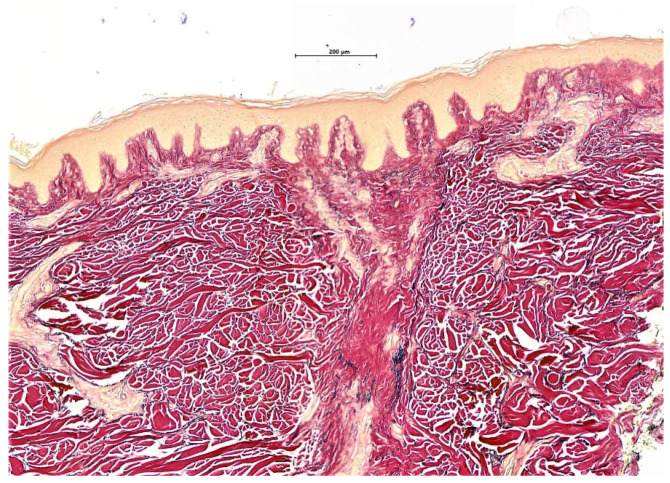
Elastic staining pattern in scarring alopecia. The elastic Van Gieson (EVG) stain highlighting a superficial wedge shaped scar of perifollicular fibrosis without significant inflammation (×40).

**Figure 6 biomedicines-09-01755-f006:**
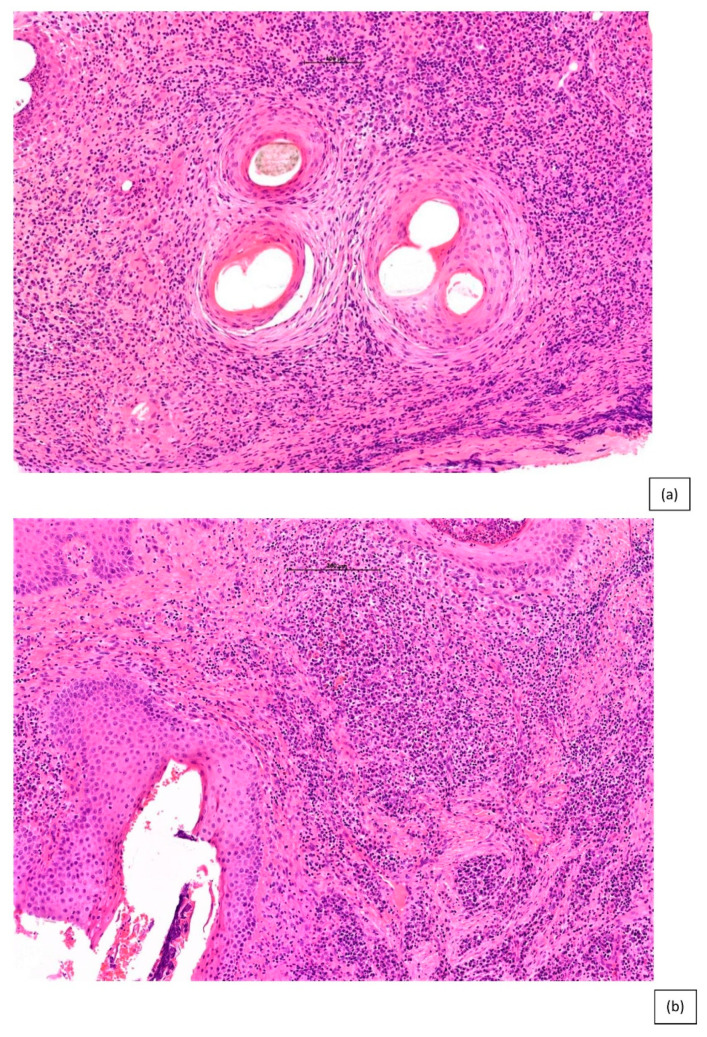
(**a**,**b**) Folliculitis decalvans. High power magnification in a horizontal section of FD with perifollicular polymorphic inflammatory cell infiltrate rich in neutrophils, lymphocytes and plasma cells (H&E ×60).

**Figure 7 biomedicines-09-01755-f007:**
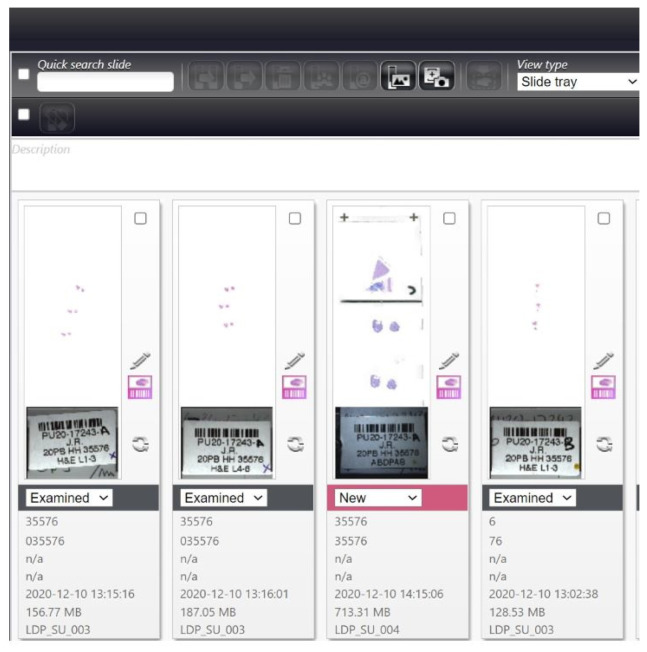
Whole Slide Imaging. A virtual slide tray of a given case with multiple horizontal and vertical sections.

**Figure 8 biomedicines-09-01755-f008:**
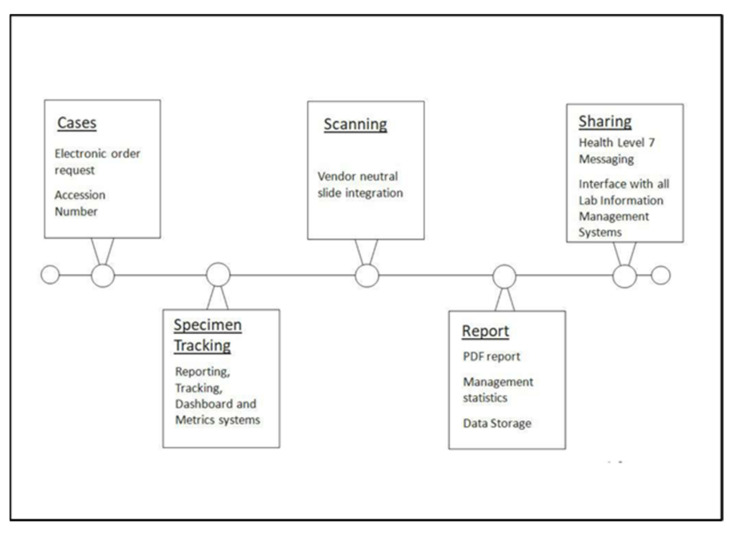
Proposed pathway. A process map of an end to end digital pathology solution.

**Table 1 biomedicines-09-01755-t001:** Primary Cicatricial Alopecia based on NAHRS classification: An overview of clinical and histological findings.

Inflammatory Infiltrate	Clinical Features	Histological Findings
Lymphocytic		
Chronic cutaneous lupus erythematosus	Erythematous scaly plaques with follicular plugging associated with hyperpigmentation and/or hypopigmentation	Follicular hyperkeratosis with ostial plugging Thickened basement membrane Vacuolar interface change with apoptotic keratinocytes along the follicular basal layer +/− at the dermo-epidermal junction between follicles Lympho-plasmacytic infiltrate involving the superficial and deep dermis—perivascular and peri-adnexal Dermal mucin Diminished sebaceous glands Widespread loss of elastin DIF shows IgG, IgM, and C3 at the BMZ
Lichen planopilaris (LPP) Classic LPP Frontal fibrosing alopecia (FFA) Graham -Little -Piccardi-Lassueur syndrome	Skin-coloured or erythematous patches of alopecia on the scalp with perifollicular erythema and perifollicular scale Progressive recession of the frontal hairline in a band-line pattern; Perifollicular erythema, follicular hyperkeratosis +/− loss of eyebrows/body hair Features of LPP with non-cicatricial alopecia of axillary and pubic hair and lichenoid follicular eruption on the trunk, limbs, face, or eyebrows	Lichenoid infiltrate around the isthmus and infundibulum Interface dermatitis Artefactual clefting between the epithelium and the stroma Premature desquamation of the inner root sheath in severely inflamed follicles Concentric perifollicular lamellar fibroplasia Wedge-shaped loss of elastin fibres Fibrosed follicular tracts
Classic pseudopelade (Brocq)	Skin-coloured scarred plaques “footprints in the snow” with minimal follicular hyperkeratosis and erythema	Reduced number of terminal hairs and loss of sebaceous glands Minimal dermal inflammation Cylindrical columns of connective tissue form at the site of former follicles Follicular stelae with no overlying follicle
Central centrifugal cicatricial alopecia	Slowly progressive expanding patch of scarring on the crown or vertex that progresses centrifugally	Concentric lamellar fibroplasia of follicles Variable lymphocytic perifollicular inflammation of level of isthmus and lower infundibulum Desquamation of inner root sheath Granulomatous inflammation and retained hair shaft fragments
Alopecia mucinosa	Grouped follicular papules, patches, and/or boggy erythematous plaques, most commonly on the head and neck	Mucin deposits in outer root sheath and subsequently the entire hair follicle Lymphocytic infiltrate
Keratosis follicularis spinulosa decalvans	Noninflammatory, follicular keratotic papules and pustules with progressive hair loss affecting scalp/eyebrows/eyelashes	Follicular plugging and hypergranulosis Fibrosis Primarily lymphocytic perifollicular infiltrate
Neutrophilic		
Folliculitis decalvans	Erythematous follicular papules, pustules and patches associated with follicular hyperkeratosis and tufted folliculitis	Interfollicular and perifollicular mixed infiltrate of neutrophils, lymphocytes, and plasma cells Marked inflammation at lower infundibulum
Dissecting cellulitis/folliculitis (perifolliculitis capitis abscedens et suffodiens)	Boggy, suppurative nodules, abscesses and sinus tracts on the vertex and posterior scalp	Follicular occlusion Early dense lymphocytic perifollicular inflammation at lower half of follicle Deep abscesses and sinus tracts of neutrophils, lymphocytes, and plasma cells in later disease
Mixed		
Acne keloidalis nuchae	Grouped follicular papules, pustules and plaques on the occipital scalp and nape of the neck with varying degrees of inflammation	Follicular occlusion Early dense lymphocytic perifollicular inflammation at lower half of follicle Deep abscesses and sinus tracts of neutrophils, lymphocytes, and plasma cells in later disease
Acne necrotica	Umbilicated, erythematous follicular papules and pustules that undergo central necrosis and resolve with varioliform scars	Perifollicular lymphocytic infiltrate in early disease Late phase changes include follicular necrosis and neutrophils in the superficial dermis
Erosive pustular dermatosis	Pustules, erosions and crusted plaques on elderly scalps	Nonspecific early findings with marked epidermal atrophy and focal erosions Chronic mixed inflammatory infiltrate and fibrosis in later lesions Not folliculo-centric

**Table 2 biomedicines-09-01755-t002:** Histological features helpful in distinguishing the most common scarring alopecias.

	LPP	CCLE	CCCA	FD
Epidermis and hair follicle epithelium	Sparing of interfollicular epidermis	Interfollicular epidermal changes (follicular plugging, vacuolar alteration and atrophy)	Eccentric atrophy of the follicular epithelium	Flattened and “squamatisation of hair follicle epithelium surrounded by a zone of fibroplasia and inflammation
Inflammation	Perifollicular infiltrates (predominately lymphocytic although histiocytes also occur) with sparing of deep vascular plexus and adnexal structures	Superficial and deep lymphocytic infiltrate involving eccrine glands	Variably dense lymphocytic perifollicular inflammation, primarily at the level of the upper isthmus and lower infundibulum	Predominantly neutrophils, with a component of both lymphocytes and plasma cells at varying depths
Mucin	Perifollicular	Interfollicular dermal mucin deposition	Perifollicular	Perifollicular
DIF	Non- specific globular IgM in Civatte bodies	Linear deposition of IgG, IgM, and C3 at the dermal–epidermal junction and follicular epithelial dermal junction	Negative	Negative
Elastic tissue staining	Loss of elastic tissue and the elastic sheath in a superficial wedge-shaped scar	Broad scar throughout the dermis and destruction of the elastic sheath surrounding the fibrous tracts	Hyalinization of the dermis with increased and thickened elastic fibres. Broad fibrous tracts with preserved elastic sheath	Superficial wedge-shaped scar with late diffuse dermal scar/fibrosis
Distinguishing features	Peri-follicular infundibular lichenoid (band-like) inflammation and apoptotic bodies	Vacuolar interface change Dermal mucin Thickened basement membrane	PDRIS found in early disease and normal appearing scalp	Polytrichia and hair shaft granulomas as a predominant feature

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
