# Peer review of "Scarring Alopecias: Pathology and an Update on Digital Developments"

_biomedicines, 2021, doi:10.3390/biomedicines9121755_

Round 1
Reviewer 1 Report
The manuscript has improved, although a missing clinical-pathology digital link still remains between the first part reviewing the well-known pathology of scarring alopecias and the second part addressing the ‘digital frontier’. Items that need attention:
- COMMENT TO REPLY 1a. The change of the TITLE reflects the two different topics discussed, ie ‘pathology’ and ‘digital developments’. But the manuscript does not address specifically the pitfalls in any dedicated PARA. The word ‘pitfalls’ should be eliminated from the title.
- INTRODUCTION: PARA 1, LINES 38-41: ‘Appropriate biopsy site selection along with close communication between the clinician and the histopathologist will improve the chance of an accurate diagnosis………..complex patients. This statement need to be referenced. Histopathology 2010, 56(1):2438. Diagnostic Pathology 2020, 26(3):114.
- PAGE TWO, PARA TWO, LINE 71: ‘Samples should be taken using at least a 4mm punch biopsy…’ Delete the word ‘at least’ from the sentence.
- PAGE TWO PARA TWO, LINES 74-76: ‘It is important to recognize that the punch biopsy device used can influence adequacy of specimen size as the internal diameter of some punch instruments can render a smaller specimen than anticipated.’ This concept has been documented in the literature and this sentence needs to be referenced: Wain M et al. Br J Dermatol 2007;156,404.
- COMMENT TO REPLY 4a:
- The statement is incorrect. Non-scarring alopecias are the real histopathological challenge to the pathologist. Scarring alopecias are not histopathologically complex, due to their diversity of histopathological aspects that can be easily detected by pathologists. This statement needs to be rephrased.
- PITFALLS (and how to avoid them) where are they specifically discussed in the manuscript? (See comment to REPLY 1a)
- COMMENT TO REPLY 10a. ‘IMF is not essential’. This is at risk of leading to the mismanagement of scarring alopecia. Routine use of IMF in scalp biopsies can highlight overlap diseases and unexpected findings in scenarios other than lupus. The statement at PAGE 3, LINES 130-132 should be rephrased to reflect a broader diagnostic approach with the use of IMF. Diagnostic Pathology 2020, 26(3):114.
- LEGENDS TO FIGURES:
- Histopathology pictures Figure 1 and Figure 2 and 5 are missing correct microscopic magnification, while others have inaccurate and discrepant magnifications. Figure 3 and Figure 4 are at different magnifications, but they are both at ‘medium power’. Need to correctly adjust the X magnifications based to the precise microscope objective used to take each photos.
- FIGURES: Figure 8. The blue squares in this diagram are too dark. Need to use a lighter background colour as the writing in the ‘proposed pathway’ blue squares is almost hidden.
Reviewer 2 Report
The revised manuscript has been reconstructed according to the reviewers’ suggestions and becomes much easier to read and understand the aim of this manuscript on histopathology of scarring alopecia with potential usefulness of digital pathology for future clinical practice.
Author Response
Many thanks for your considered response.
Best wishes,
Dr. Donna Cummins
on behalf of Dr. Iskander Chaudhry & Dr. Matthew Harries
Round 2
Reviewer 1 Report
The authors have attended to this reviewer's comments introducing the appropriate references where needed.
There are still a few technical items to correct:
- Figure 3 and Figure 4 have been listed as having the same magnification of X40. This is incorrect, as Figure 3 has been clearly been taken at a higher magnification than Figure 4. This needs to be amended.
- Remove the word 'magnification' in each legend: it is not necessary. Each legend should read for example: Figure 3.....(H&E X40)
- The X in the legends should be a capital letter and not in lower case.
Author Response
Dear Reviewer,
Many thanks for your comments. Please find updated figure legends in the draft attached.
Best wishes,
Dr. Donna Cummins
Round 3
Reviewer 1 Report
This review V3 does not address the discrepancy in magnification size of the histopathology pictures taken.
This reviewer believes that Figure 3 and 4 are still incorrect, and not matching the correct size of the magnification. This is the more evident discrepancy but it likely affects all the pathology pictures in the manuscript.
Please review all the histology pictures with the pathologist who took the photos and who will need to review all the magnifications and correct the magnification numbers to include the microscope correction factor of X10 (should be: objective size used multiplied by 10. For example a picture taken with at 4X objective will be X40; A picture taken with objective 10X will be X100 etc) This reviewer is unfamiliar with X5 as in legend to Figure 1. What was the objective used to take this photograph?
Author Response
Dear Reviewer,
Many thanks for your considered comments.
On review, we are satisfied that the image magnifications are correct. The figures represent digital magnifications and it is not possible to judge this according to standard microscopy objective magnifications. Example magnification is attached which equates to 4.62x magnification.
Kind regards,
Dr. Donna Cummins

Round 4
Reviewer 1 Report
unfortunately, this reviewer disagrees with the authors comments to the magnifications of the pathology pictures. Particularly as they do not reflect the size observed.
However, this manuscript is intended to a broader readership, which likely will not notice the discrepancy.
This manuscript is a resubmission of an earlier submission. The following is a list of the peer review reports and author responses from that submission.
Round 1
Reviewer 1 Report
This manuscript is a review of scarring alopecias outlining the benefits of the practical and diagnostic use of digital pathology.
GENERAL COMMENTS
The manuscipt is well-written but appears unbalanced with respect to the Title of the which misleads the reader's expectations. The title "....Digital Pathology Solution supporting the investigation..." lends the reader to believe there is a major involvement of Digital pathology in the diagnostic aspects of scarring alopecias.
While paragraphs 1-5 and 11-14 on digital pathology and relevant aspects of tertiary specialist alopecia services are relevant and novel, Paragraphs 6-10.6 (Cicatricial alopecias) and Tables I and II are disconnected (as they are well known histopathology, only). They do not show any digital-histopathology correlation, as the title of the manuscript would imply.
To improve this manuscript, the authors should link the two main elements of this manuscript (histopathology + digital pathology) and comment on the morphological observations (commonalities and differences) for each scarring alopecia entity between the standard microscopic approach to the diagnosis with the advantages (or not) derived from the diagnostic approach with digital pathology.
QUESTION: Why were the non-scarring alopecias included and only PCAs described?
SPECIFIC COMMENTS
PARAGRAPHS 6-10.5. The histopathology sections describe well-known findings of each entity. This part is too long and not focused on the true novelty of this paper of digital pathology. Please see above further comments.
PARA 7, SCALP BIOPSY, SITE SELECTION, AND OPTIMIZING SAMPLE QUALITY, Lines 145-149.
This comment is inaccurate and misleading. The following referenced statement (11) " if only a single biopsy is obtained, .......vertical rather than horizontal sectioning should be prioritized" albeit referenced, in routine practice, and it is not actioned.
If a single punch biopsy is received, horizontal sectioning can answer all the queries. Moreover, horizontal sections ensure a better assessment of all the hair follicles within the punch biopsy (versus only a few visualized in the vertical sections) and enable the pathologist to comment on the degree of inflammation which in turn will help the clinician understand if the process of scarring alopecia is still active or burnt out (provided an accurate 'active' biopsy site was chosen). If need be, after the assessment on horizontal sections, and if the epidermis is still not seen on deeper horizontal sections, the specimen can be reoriented vertically for further examination.
PARA 8, SAMPLE PROCESSING AND PATHOLOGY REPORTING, Lines 166-172
Whilst the vertical sections may give 'a quick' answer confirming a scarring process, it is only the horizontal sectioning which will broaden the assessment and be more informative on the aetiology of the ongoing process leading to scarring alopecia. The statement " ....no real diagnostic value" is inaccurate. Follicular density and hair counts may apparently be less relevant in the setting of a scarring alopecia. However, horizontal sections ensure assessment of all the hair follicles with the degree of inflammatory cell infiltrate and enlighten on the phase of 'activity' of the scarring process (as commented above). Horizontal sections allow also for the assessment of concomitant pathologies within the same biopsy (multifactorial alopecia), as both scarring and non-scarring processes (with miniaturization) may be present simultaneously (J Cutan Pathol 2016;Jun43(6):483-491).
PARA 8, SAMPLE PROCESSING AND PATHOLOGY REPORTING, Lines 174-175
"The routine use of special stains is not recommended......." This referenced comment (14) is misleading, and does not reflect the reality of common practice. The standard use of special stains (VVG, PAS, Colloidal iron) offers the trained trichopathologist additional information for completion of the assessment of both early (and late) scarring as well as of non-scarring alopecia processes.
PARA 8, SAMPLE PROCESSING AND PATHOLOGY REPORTING, Lines 183-185
Should be added that colloidal iron is also helpful in identifying the perifollicular mucinous component of early scarring alopecia as initially described in LPP (J Am Acad Dermatol 2007;57(1):47-53) but observed also in all types of early scarring alopecia. Table II is therefore inaccurate, as mucin may be seen in the perifollicular scarring of all the listed entities (LPP, CCLE, CCCA, FD).
Toluidine blue is a valuable stain helpful in identifying intrafollicular/epithelial mucin, as in follicular mucinosis and folliculotropic CTCL. It is commonly paired with PAS, so both stains are present in the same slide.
PARA 8, SAMPLE PROCESSING AND PATHOLOGY REPORTING, Lines 188-190
The following comment is inaccurate and misleading.
" However, DIF is usually unnecessary in differentiating the majority of PCA cases histologically, except in inconclusive cases where lupus is questioned".
The role of DIF in completing the assessment of scarring alopecia is broader than herein described and is paramount in excluding other causes of scarring alopecia. (Diagnostic Histopathology 2020;26(3):114-127).
PARA 13.3 ARTIFICIAL INTELLIGENCE AND HAIR DISEASES Lines 465-466
The standard practice of biopsy taking is the use of a 4mm punch. Almost all biopsies reviewed by this reviewer are received in this size. The comment "biopsy diameter varies from case to case depending on the biopsy taker" while it may be true, it is a rare event, and leads the reader to believe that punches of different sizes are received in the laboratory much more frequently than is truly the case. This comment should be rephrased to reflect this.
Reviewer 2 Report
This manuscript seems to have major drawbacks. It includes both (1)review part on the general pathological clinical challenges and prospects of introducing technologies such as digital pathology (DP) and AI into routine clinical pathology, and (2) the observational description part of a very practical classification of the specific diagnostic process for scarring hair loss (PCA), therefore potential readers would be confused and the purpose of this manuscript become unclear.
From” 1. Introduction” to “4. Artificial Intelligence interface”, there is no novel informative description and no information about PCA can be obtained.
Moreover, Sub sections 5-7 contain descriptions specific toPCA, but subjectively the authors conclude that the skilled technicians are essential for correct pathological diagnosis, despite mentioning the use of DP and AI.
From “11. Role of CPC” to “14. Conclusion”, again, the descriptive, redundant statement of future prospects came back, and only in “12. Digital Pathology in Practice”, the results of authors’ own clinical practice are briefly summarized.
I strongly recommend the authors re-consider the construction of the revised manuscript more focused on practical and appropriate PCA diagnosis procedure utilizing digital-clinical technology.